# OpenReview forum: "Character-LLM: A Trainable Agent for Role-Playing"
_EMNLP/2023/Conference — EMNLP 2023 Main_

### Official Review · Reviewer_fhYt · 2023-08-01

**Soundness:** 3

**Excitement:**

4: Strong: This paper deepens the understanding of some phenomenon or lowers the barriers to an existing research direction.

**Paper Topic And Main Contributions:**

This paper investigates how to simulate a specific person with large language models (LLMs). Different from previous works that prompt existing LLM, the authors construct the experience of that person and then fine-tune an LLM on the experience dataset. The experience consists of (1) character experience generated by another LLM based on the profile, and (2) protective experience that shows the person has no idea about unrelated knowledge. Evaluated by GPT 3.5, the trained model performs better than Alpaca, Vicuna, and is comparable with ChatGPT, in several aspects such as memorizing the character profile and reducing hallucination.

**Questions For The Authors:**

See above

**Reasons To Accept:**

1. This paper proposes an experience reconstruction method to extend a character's profile to many detailed scenes that mimic interactions between the target character and other characters. The experience can provide more information for character simulation. This method may inspire future work on character data construction.
2. This paper proposes constructing and using protective experience to reduce hallucination during role-playing. Trained on protective experience, the model can pretend to be ignorant of the knowledge that the character does not know. The character hallucination problem is important, and this work gives some insight.

**Reasons To Reject:**

1. Lack of important details. It's unclear which LLM did the authors use for scene extraction and experience generation (both character experience and protective experience). Does the selection of this model important? Does the generated experience faithful and without hallucination? How many scenes are there in the protective experience for training? For evaluation, how are the questions/topics selected?
2. Lack of ablation study to show the superiority of the proposed experience construction method. I think adding a comparison between models fine-tuned on different kinds of character data helps.

**Reproducibility:**

2: Would be hard pressed to reproduce the results. The contribution depends on data that are simply not available outside the author's institution or consortium; not enough details are provided.

**Reviewer Confidence:**

3: Pretty sure, but there's a chance I missed something. Although I have a good feel for this area in general, I did not carefully check the paper's details, e.g., the math, experimental design, or novelty.

---

> ### Author Rebuttal · Authors · 2023-08-28
>
> We sincerely thank you for your constructive feedback and kind evaluation. We have made a substantial effort in responding to your questions to enhance clarity.
>
> **Q1: Lack of important details.**
>
> **A1:** To ensure clarity and reproducibility, we will add more details of implementations in the revision and open source to complete the experiment pipeline and results. Here is the outline:
>
> 1) Code and prompts for experience reconstruction;
> 2) Experience dataset to train the character simulacra;
> 3) Code and hyper-parameters for training the agents;
> 4) Generation results of different models (including our trained agents and baselines);
> 5) Model weights of our trained agents.
>
> To address your concerns, we are using OpenAI's gpt-3.5-turbo with temperature=0.7 for the whole experience reconstruction part, including scene extraction, experience generation and protective experience construction. The prompts used for experience reconstruction can be found in the Appendix. The training details can be found in Section 4.2 in the submission.
>
> **Q2: Does the selection of models for experience reconstruction important?**
>
> **A2:** Thank you for raising this great question. It is straightforward that the more powerful of the LLM is, the better the quality of data it can generate. However, we believe that the capability of LLMs is not the primary bottleneck when it comes to constructing the training corpus for character simulacra. Based on comparisons with models distilled from more data and more powerful LLMs, such as Alpaca 7B (which obtains 52K training data from ChatGPT) and Vicuna 7B (trained on ShareGPT with high-quality responses generated by ChatGPT and GPT-4), the proposed experience reconstruction framework enables our trained agents to achieve comparable performance with very little training data.
>
> **Q3: Does the generated experience faithful and without hallucination?**
>
> **A3:** This is a great question. During the experience reconstruction of historical figures, some scenes or utterances can be hallucinated by large language models. However, our primary focus is not on identifying these hallucinations unless they impact the quality of character simulation, such as inaccuracies in personalities, values, and character memories. We emphasize that Character Hallucinations are the main focus of our work. Distinct from conventional hallucinations in large language models, character hallucinations occur when agents deviate from the character and exhibit unrelated knowledge or abilities of the simulated individual. Exploring this phenomenon further is a direction we intend to pursue in the future.
>
> **Q4: How many scenes are there in the protective experience for training?**
>
> **A4:** We find it can be very effective to add just a few protective experiences in our experiments. The protective experience consists of a 10%-20% proportion of the training data. We plan to include more statistics about the dataset in the revision and open-source the complete experiment pipeline and results.
>
> **Q5: For evaluation, how are the questions/topics selected?**
>
> **A5:** Our interview questions consist of more than 100 different questions per character, for nearly 1000 questions in total. We construct these questions with the assistance of ChatGPT. To make the questions diverse and cover all the aspects that we would like to evaluate on the agents, we enumerated various topics and prompted ChatGPT to write interview questions based on these topics. We manually examined interview questions of one character and found that ChatGPT can generate high-quality interview questions. Some questions are easy to answer like "Where are you come from?", while some questions require a deep understanding of the characters, e.g. one question for Caesar: "Have you ever regretted a decision you've made as a leader?". By interviewing with these questions, we are able to reliably evaluate the agents' performance on the personal knowledge, preferences, values, personalities and stability of Character Simulation. To further enrich the clarity, we will add detailed statistics and visualization on interview topics and questions in the revised version.
>
> **Q6: Lack of ablation study to show the superiority of the proposed experience construction method. I think adding a comparison between models fine-tuned on different kinds of character data helps.**
>
> **A6:** We agree that it would be helpful to compare our method with models directly fine-tuned on character data. However, we did not add this comparison in the first place because we do not believe it will work. We collect Wikipedia pages as the profiles for experience reconstruction. Without our method, these Wikipedia pages are the only data that is related to the characters. We believe directly fine-tuning language models on these documents leads to poor performance, because
> 1) The amount of training data is limited;
> 2) There is only a brief introduction about the characters, in which many details are omitted;
> 3) The text format is not consistent with the interaction style of agents.
>
> Regardless of these complications, we will add this comparison in the revision to show the effectiveness of our method.

---

### Official Review · Reviewer_3D9c · 2023-08-05

**Soundness:** 3

**Excitement:**

2: Mediocre: This paper makes marginal contributions (vs non-contemporaneous work), so I would rather not see it in the conference.

**Paper Topic And Main Contributions:**

The paper presents a novel approach for training trainable agents to simulate specific individuals by providing them with profiles and experiences. The authors propose a framework that includes experience reconstruction, experience upload, and protective experiences. They evaluate the trained agents through interviews and compare them with baseline models. The results show that the trainable agents outperform the baselines in terms of personality, memorization, hallucination, and stability. The paper provides valuable insights into building better character simulacra.


**Questions For The Authors:**

(1) Could you please provide more details about the implementation of the trainable agents, such as the specific architecture and training settings used?
(2) Have you considered comparing the trained agents with more baselines, such as other instruction-tuned models or models trained with different methodologies?
(3) Can you discuss the potential ethical concerns of using trainable agents and how you plan to address them?

**Reasons To Accept:**

(1) The paper introduces a novel approach for training trainable agents to simulate specific individuals, which is a step closer to character simulacra.
(2) The framework proposed by the authors, including experience reconstruction, experience upload, and protective experiences, is well-structured and practical.
(3) The evaluation process, including interviews and AI-based judging, provides comprehensive insights into the performance of the trained agents.

**Reasons To Reject:**

(1) The paper lacks implementation details, such as the specific architecture and training settings used for the trainable agents.
(2) The evaluation could benefit from comparisons with more baselines and a larger number of interview questions.
(3) The paper could provide more in-depth discussions on the limitations and potential ethical concerns of using trainable agents.

**Reproducibility:**

2: Would be hard pressed to reproduce the results. The contribution depends on data that are simply not available outside the author's institution or consortium; not enough details are provided.

**Reviewer Confidence:**

3: Pretty sure, but there's a chance I missed something. Although I have a good feel for this area in general, I did not carefully check the paper's details, e.g., the math, experimental design, or novelty.

---

> ### Author Rebuttal · Authors · 2023-08-28
>
> Thank you very much for providing us with very constructive comments.  We have addressed your concerns in detail below.
>
> **Q1: Could you please provide more details about the implementation of the trainable agents, such as the specific architecture and training settings used?**
>
> **A1:** To ensure clarity and reproducibility, we will add more details of implementations in the revision and open source to complete the experiment pipeline and results. Here is the outline:
> 1) Code and prompts for experience reconstruction;
> 2) Experience dataset to train the character simulacra;
> 3) Code and hyper-parameters for training the agents;
> 4) Generation results of different models (including our trained agents and baselines);
> 5) Model weights of our trained agents.
>
> To address your concerns, we utilized LLaMA 7B to initialize the model and train on the experience reconstruction data for our agents, without modifying the LLaMA architecture. For each agent, we trained with AdamW optimizer for 10 epochs, with the peak learning rate being 2e-5, the maximum length being 2048 and the batch size being 64. The rest of the training details can be found in Section 4.2 in the submission.
>
> **Q2: Have you considered comparing the trained agents with more baselines, such as other instruction-tuned models or models trained with different methodologies?**
>
> **A2:** This is a constructive comment. We recognize that a thorough comparison is essential and have chosen well-established instruction-tuning models as baselines.  These models have been trained using various methodologies, including SFT (Alpaca 7B and Vicuna 7B) and RLHF (ChatGPT). We believe these baselines are sufficient to reflect the effectiveness of the proposed framework. We believe these baselines are sufficient to reflect the effectiveness of the proposed framework. However, to further enrich the comparisons with models having different backbones, we will include additional baselines such as LLaMA 2-Chat and Vicuna 13B in the revised version.
>
> **Q3: 3. About more interview questions**
>
> **A3:** Our interview questions consist of more than 100 different questions per character, for nearly 1000 questions in total. We construct these questions with the assistance of ChatGPT. To ensure diverse coverage and encompass all the aspects we wish to evaluate in the agents, we cataloged various topics and prompted ChatGPT to create interview questions based on these subjects. Upon manual review of the interview questions for one character, we discovered that ChatGPT can generate high-quality interview questions. Some questions are easy to answer, such as "Where do you come from?" While others demand a profound comprehension of the characters. For instance, one question for Caesar is: "Have you ever regretted a decision you've made as a leader?" Through these questions, we can consistently evaluate the agents' performance in terms of personal knowledge, preferences, values, personalities, and the stability of Character Simulation. We will add detailed statistics and visualization on interview topics and questions. Moreover, we will add more interview questions in the revised version.
>
> **Q4: Can you discuss the potential ethical concerns of using trainable agents and how you plan to address them?**
>
> **A4:** We recognize the importance of clarifying the limitations and ethical concerns surrounding the use of trainable agents. We intend to include discussion sections covering the following points:
>
> **Limitations of Using Trainable Agents**
>
> - Modality: Trainable agents rely on language models, allowing them to manipulate only textual content. Consequently, they are limited in their applicability to tasks that require multi-modal inputs or outputs, e.g. video games and voice assistants.
> - Cost: Since the agents are character-specific, training and deploying a large number of agents to simulate different characters can be both time-consuming and resource-intensive.
> - Editability: Trainable agents may encounter difficulty in adapting to learning new knowledge and skills without further training.
> - Control:  It would be challenging to predict the action agents will take during the interactions, especially in scenarios demanding controlled generated outcomes (e.g., scripted plotlines in stories).
> - Accuracy: Trainable agents might lack the capability to fully emulate the behavior of a given character, and accurately identifying divergent actions can be non-trivial.
>
> **Ethical Concerns of Using Trainable Agents**
>
> - Privacy and Exploitation: Agents could be trained on private or personally identifiable data to simulate specific individuals. For instance, agents might function as user simulators for personalized recommendations. The use of agents to simulate humans raises potential risks. Just as techniques generating fake videos, photos, or voices have ethical implications, trainable agents might generate fabricated texts attributed to targeted individuals. Consequently, constraints on the data usage of trainable agents and rigorous censorship are essential. Moreover, the Data providers should be informed about and give consent for the use of their personal data. In this paper, we utilize publicly available data from Wikipedia to construct the training data and focus on simulating historical and fictional characters to avoid this concern.  Additionally, these trainable agents are strictly designated for research purposes.
> - Harmful Content: There is a risk that agents might be trained to produce harmful or toxic texts using toxic experiential data. To address this, developing tools for toxicity detection is imperative. Moreover, careful alignment of pre-trained large language models is necessary to mitigate and identify the generation of toxic data. By using the safely aligned LLM (gpt-3.5-turbo) for experience reconstruction and the backbone model that is pre-trained on filtered data (LLaMA), we mitigate the toxicity in the trainable agents.
> - Misunderstanding: When employing agents to simulate characters from diverse cultures or historical periods, cultural discrepancies may arise. Trainable agents might misinterpret local culture or ethical norms, leading to subpar performance or even offending local users. Thus, constructing high-quality experiential training and evaluation data is critical. We utilize "protective experience" to eliminate unrelated knowledge in agents. By rejecting queries that are beyond the character's knowledge, misinterpretation can be mitigated.

---

### Official Review · Reviewer_r5Uk · 2023-08-05

**Typos Grammar Style And Presentation Improvements:** L447 OpenAI
**Soundness:** 4

**Excitement:**

4: Strong: This paper deepens the understanding of some phenomenon or lowers the barriers to an existing research direction.

**Missing References:**

SODA: Million-scale Dialogue Distillation with Social Commonsense Contextualization - Kim et al.

**Paper Topic And Main Contributions:**

This paper introduces an innovative pipeline designed to construct trainable agents on LLMs that can convincingly emulate specific historical figures, such as Beethoven. The pipeline commences with the collection of comprehensive profiles for the chosen characters. These profiles are then utilized to prompt the LLMs to generate detailed scenes and experiences that are characteristic of these figures. Subsequently, a specialized LLM is fine-tuned based on these character-specific experiences, resulting in a convincing simulacrum of the character. Experimental evaluations, conducted using ChatGPT as a benchmark, demonstrate that the performance of these character simulacra is comparable to ChatGPT.

**Questions For The Authors:**

1. In your experiments with RLHF-trained or distilled from RLHF models such as ChatGPT and Vicuna, have you noticed them generating safe responses like "As an AI, I can't..."? If so, how do you handle these situations in your evaluation?

2. Do you notice a preference for ChatGPT agents evaluated by humans since you use GPT as the evaluator?

3. Do you observe hallucinations in scenes? If so, how do you prevent them?


**Reasons To Accept:**

1. The paper introduces a novel pipeline for creating trainable agents on LLMs that can emulate specific characters. This approach is innovative and could have wide-ranging applications in various fields, such as virtual assistants, education, and entertainment.

2. This paper presents a thorough process that covers everything from gathering character profiles to refining LLMs using those profiles. This meticulous method could serve as a guide for other scholars in the same field.

3. The paper also discusses potential flaws and areas for improvement in the proposed approach, indicating a thorough and thoughtful analysis.

**Reasons To Reject:**

1. While the Protective Experience method is used to reduce hallucination issues, there is a possibility that it may result in bias or incorrect information being incorporated into the agent. This is a matter of concern for future usage.

2. Including a human evaluation could potentially increase its credibility.

**Reproducibility:**

4: Could mostly reproduce the results, but there may be some variation because of sample variance or minor variations in their interpretation of the protocol or method.

**Reviewer Confidence:**

4: Quite sure. I tried to check the important points carefully. It's unlikely, though conceivable, that I missed something that should affect my ratings.

---

> ### Author Rebuttal · Authors · 2023-08-28
>
> We appreciate your thorough evaluation of our work and your constructive feedback.  We are grateful for your insights and agree that addressing the highlighted points can significantly enhance the clarity and appeal of our paper.
>
> **Q1: While the Protective Experience method is used to reduce hallucination issues, there is a possibility that it may result in bias or incorrect information being incorporated into the agent. This is a matter of concern for future usage.**
>
> **A1:** This is an insightful question. Bias and incorrect information may be introduced when training an agent, because the target characters may have their intrinsic biases and the pre-trained backbone models may carry such bias and incorrect hallucinations in their weights. Thus, it can be difficult to identify when and how this bias and misleading information is introduced. Furthermore, bias and incorrect knowledge are not our primary concerns when building high-quality character simulacra. Some bias may be intentionally introduced into the trainable agents to simulate the corresponding characters, who harbor their own biases based on their life experiences.
> Moreover, the "protective experience" does not cause bias and incorrect information. Instead, protective experience mitigates the negative effects caused by incorrectness, by helping the agents forget unrelated knowledge. We will add a discussion about this in the revision.
>
> **Q2: Including a human evaluation could potentially increase its credibility.**
>
> **A2:** Your suggestion is essential. Human evaluation is a common practice in generation tasks, in which the metrics are hard to define and evaluated by accuracy. However, we emphasize the difficulties of human evaluation of Character Simulacra. During our preliminary small-scope human evaluation, we found that regular labelers do not acquire enough knowledge of the given characters and have different biases towards them, which leads to poor evaluation results and large variances. We intend to focus on building a more reliable evaluation methodology in our future work.
>
> **Q3: In your experiments with RLHF-trained or distilled from RLHF models such as ChatGPT and Vicuna, have you noticed them generating safe responses like "As an AI, I can't..."? If so, how do you handle these situations in your evaluation?**
>
> **A3:** This is a great comment. We find there are very few responses including text like “As an AI language model ....” during the evaluation. ChatGPT as the evaluator accurately recognizes such responses and assigns low scores. Moreover, we find Vicuna triggers the most safety responses (3 times out of 100+ interviews for a character) while our trained agents do not trigger any. We believe the general RLHF alignment for harmlessness affects the role-playing ability of agents.
>
> **Q4: Do you notice a preference for ChatGPT agents evaluated by humans since you use GPT as the evaluator?**
>
> **A4:** This is an insightful comment. During our early experiments, we randomly checked 15 interviews per character (135 interviews in total) generated by different models. We find that the ChatGPT agents are slightly better than our trained agents, which is consistent with the results of the ChatGPT evaluator. And it motivates us to conduct the evaluation using GPTs. We hypothesize that ChatGPT as the evaluator prefers its own generated texts to some extent but it affects little to the evaluation results. To ensure clarity, additional evaluation will be conducted using Claude in the revision, which is another powerful LLM that is independent of ChatGPT. We intend to focus on building a more reliable evaluation methodology in our future work.
>
> **Q5: Do you observe hallucinations in scenes? If so, how do you prevent them?**
>
> **A5:** Thank you for raising this question. During the experience reconstruction of historical figures, some scenes or utterances can be hallucinated by large language models. However, our primary focus is not on identifying these hallucinations unless they impact the quality of character simulation, such as inaccuracies in personalities, values, and character memories.  We emphasize that Character Hallucinations are the main focus of our work.
> Distinct from conventional hallucinations in large language models, character hallucinations occur when agents deviate from the character and exhibit unrelated knowledge or abilities of the simulated individual. Exploring this phenomenon further is a direction we intend to pursue in the future.
>
> **Q6: About Missing References & Typos**
>
> **A6:** We've added the missing references and amended the typos in the revision. Your keen observation is indeed valuable.

---

### Meta-Review · Area_Chair_ZcRd · 2023-09-12

**Recommendation:** 4

**Metareview:**

Overall, the paper addresses an intriguing subject of how characters can be simulated using LMs. The authors devise a novel methodology and playground for making progress on this that allows them to train agents that they then evaluate for their memories/experiences.

The reviewers jointly all recognize the novelty, thoroughness and insightfulness of many aspects of the work. However, they do also highlight that aspects of rigor (e.g. baselines/ablations, human evals, many important implementation details) could be strengthened. Finally, for this work in particular, I do think there are several ethical concerns with reviewers disagreeing on the extent to which authors address this. If accepted, I would like to see the authors strengthen this aspect based on my own brief reading of the paper. I also encourage the authors to strengthen the implementation details, as I do find the author rebuttals to not adequately address these matters, though the authors do make extensive promises in this vein.

With all of this said, I believe the paper breaks new ground and some of its clear limitations can be addressed as this line of work grows. If the questions of rigor and ethical concern can be addressed, this paper could play a formative role in imaging new possibilities using language models.

---

### Decision · Program_Chairs · 2023-10-07

**Decision:**

Accept-Main

**Comment:**

Overall, the paper addresses an intriguing subject of how characters can be simulated using LMs. The authors devise a novel methodology and playground for making progress on this that allows them to train agents that they then evaluate for their memories/experiences.

The reviewers jointly all recognize the novelty, thoroughness and insightfulness of many aspects of the work. However, they do also highlight that aspects of rigor (e.g. baselines/ablations, human evals, many important implementation details) could be strengthened. Finally, for this work in particular, I do think there are several ethical concerns with reviewers disagreeing on the extent to which authors address this. If accepted, I would like to see the authors strengthen this aspect based on my own brief reading of the paper. I also encourage the authors to strengthen the implementation details, as I do find the author rebuttals to not adequately address these matters, though the authors do make extensive promises in this vein.

With all of this said, I believe the paper breaks new ground and some of its clear limitations can be addressed as this line of work grows. If the questions of rigor and ethical concern can be addressed, this paper could play a formative role in imaging new possibilities using language models.